# Odorants of *Capsicum* spp. Dried Fruits as Candidate Attractants for *Lasioderma serricorne* F. (Coleoptera: Anobiidae)

**DOI:** 10.3390/insects12010061

**Published:** 2021-01-12

**Authors:** Salvatore Guarino, Sara Basile, Mokhtar Abdulsattar Arif, Barbara Manachini, Ezio Peri

**Affiliations:** 1Institute of Biosciences and Bioresources (IBBR), National Research Council of Italy (CNR), Corso Calatafimi 414, 90129 Palermo, Italy; salvatore.guarino@ibbr.cnr.it; 2Department of Agricultural, Food and Forest Sciences (SAAF), University of Palermo, Viale delle Scienze, Building 5, 90128 Palermo, Italy; sara.basile92@virgilio.it (S.B.); mokhtar.a.arif@gmail.com (M.A.A.); ezio.peri@unipa.it (E.P.); 3Plant Protection Directorate, Ministry of Agriculture, Abu-Ghraib 10081, Baghdad, Iraq

**Keywords:** α-ionone, β-ionone, cigarette beetle, *Capsicum annuum*, *Capsicum frutescens*, *Capsicum chinense*

## Abstract

**Simple Summary:**

The cigarette beetle, *Lasioderma serricorne* F. (Coleoptera: Anobiidae), is an important pest of stored products. It can be monitored using pheromone traps with or without a food source as a synergistic attractant. The study objective was to assess the response of *L. serricorne* to volatile organic compounds (VOCs) from different chili fruit powders to identify semiochemicals involved in this attraction which could be used as synthetic co-attractants in pheromone traps. Olfactometer results indicated that *Capsicum annuum* and *C. frutescens* VOCs elicit attraction toward *L. serricorne* adults, while *C. chinense* VOCs elicit no attraction. Chemical analysis and behavioral assays indicated a primary role for polar compounds in the attraction toward these sources. α-Ionone and β-ionone, compounds abundant in the attractive species and which elicited positive results in the olfactometer bioassays, may be promising candidates as attractant and/or pheromone synergist in monitoring traps for *L. serricorne*.

**Abstract:**

The cigarette beetle, *Lasioderma serricorne* F. (Coleoptera: Anobiidae) is an important food storage pest affecting the tobacco industry and is increasingly impacting museums and herbaria. Monitoring methods make use of pheromone traps which can be implemented using chili fruit powder. The objective of this study was to assess the response of *L. serricorne* to the volatile organic compounds (VOCs) from different chili powders in order to identify the main semiochemicals involved in this attraction. Volatiles emitted by *Capsicum annuum*, *C. frutescens,* and *C. chinense* dried fruit powders were tested in an olfactometer and collected and analyzed using SPME and GC-MS. Results indicated that *C. annuum* and *C. frutescens* VOCs elicit attraction toward *L. serricorne* adults in olfactometer, while *C. chinense* VOCs elicit no attraction. Chemicals analysis showed a higher presence of polar compounds in the VOCs of *C. annuum* and *C. frutescens* compared to *C. chinense*, with α-ionone and β-ionone being more abundant in the attractive species. Further olfactometer bioassays indicated that both α-ionone and β-ionone elicit attraction, suggesting that these compounds are candidates as synergistic attractants in pheromone monitoring traps for *L. serricorne*.

## 1. Introduction

The cosmopolitan cigarette beetle, *Lasioderma serricorne* (F.) (Coleoptera: Anobiidae), is one of the most damaging pests in food storage and the tobacco industry worldwide [1,2,3,4,5] and is of increasing importance for damage inflicted at heritage sites such as museums [6,7] and herbaria [8,9]. Adult beetles penetrate packaged commodities, leaving a neat round hole, which classifies this species in the category of “true penetrators” of food packages [10,11]. The majority of the damage is caused by larval feeding resulting in direct loss of the product and by a reduction of the market value of the products due to the presence of dead insects, frass, exuviae, and gnawed particles [12,13].

Standard control methods for *L. serricorne* use chemical treatments which have negative environmental consequences, carry risks to the workforce, and have a negative impact on consumer health. Moreover, they are becoming less effective as the insect develops resistance [14,15,16,17,18] and the increasing regulatory restrictions on the use of insecticides is necessitating the adoption of alternative control methodologies [19,20,21]. The implementation of new and alternative methods could lead to a holistic integrated pest management (IPM) program, which is strongly desired by the manufacturers [19,20,21,22]. In this context, the use of pheromones and other semiochemicals, such as food attractants (kairomones), for monitoring and control purposes is highly desirable for developing bio-rational control methods [23,24,25].

Female *L. serricorne* releases a sex pheromone whose main component is (4S,6S,7S)-4,6-dimethyl-7-hydroxynonan-3-one, commonly named serricornin [26,27]. This molecule is used with sticky traps for monitoring *L. serricorne* [9,28,29,30] and strongly attracts male beetles. To optimize attraction, the pheromone could be used with synergistic plant-derived volatiles that mimic food and oviposition sites [13]. In particular, adult cigarette beetles are attracted by different VOCs emitted by plant-derived substrates such as cured tobacco leaves, coffee beans, black tea leaves, cocoa powder, wheat flour, soybean flour, corn, chili, cayenne pepper, paprika, and coriander seeds [31,32,33]. The beetles oviposit on roasted coffee beans, cocoa powder, black and green tea leaves, and unpolished rice [31]. Females respond more sensitively to plant volatiles than males [31,33], which may be due to their necessity to locate food sources and oviposition sites [13].

The chili powder of *Capsicum annuum* L. has been reported as a promising attraction source for *L. serricorne* resulted in laboratory studies [33]. Moreover a recent study from Guarino et al. [9] proved that this can be effective in increasing the pheromone traps captures in an infested herbarium. However, the volatiles responsible for this attraction have not been evaluated.

The study objective was to characterize the volatile cues involved in the attraction of *L. serricorne* towards chili powder in order to identify semiochemical(s) that could be used as co-attractants for pheromone baited traps.

The specific objectives were:Test the attractiveness of VOCs released by dried fruits belonging to the genus *Capsicum* to identify species that elicit major attraction to *L. serricorne* adults.To prepare active extracts of one or more attractive species.To use bioassay-driven fractionation to characterize the attractant(s) present in extracts from the attractive species.To test singly the most abundant chemicals of the active fraction.

## 2. Materials and Methods

### 2.1. Insect

A *L. serricorne* colony was obtained from infested substrates and reared in an environmentally controlled room (24 ± 2 °C, 70 ± 10% RH, photoperiod 16L:8D), in plastic cages (750 mL volume) with the top punctured with mesh-covered holes for ventilation. Each plastic cage contained a rearing substrate made of a mixture containing chamomile powder (100 g); 00 flour (95 g); 0 flour (60 g); bran (40 g); and brewer’s yeast (5 g). Adults for the experiments were randomly chosen from the rearing cages, isolated individually in 15 mL vials, and placed in the olfactometer room for acclimatization one hour before the behavioral bioassays.

### 2.2. Bioassays with Capsicum Fruit Powders

Bioassays were carried out to test the beetle response to the powder obtained from the dried fruits of three different *Capsicum* species, *C. annuum*, *C. frutescens* L., and *C. chinense* Jacquin, grown at the Palermo botanical garden using organic farming methods. The response of beetles to odors was investigated using a Y-tube olfactometer made of a central polycarbonate body (stem 9 cm; arms 8 cm at 130 angle; ID 1.5 cm) sandwiched between two glass plates [34]. A stream of medical-grade compressed air, humidified by bubbling through a water jar, was regulated in each arm by a flowmeter at 0.2 L min^−1^. Tests were conducted from 8:30 to 14:00 h and the temperature in the bioassay room was 24 ± 1 °C. At the beginning of the bioassays, stimuli were randomly assigned to each arm, placed in 20 mL glass jars where the air was passing through. Stimuli were replaced after ten trials as described below, and their position was reversed after five replicates in order to minimize any possible bias. At every switch of the stimulus position, the glass parts of Y-tube olfactometer were changed and the body walls were cleaned with acetone. At the end of the experiments, the glass parts were cleaned with acetone and distilled water, and baked overnight at 200 °C, while the polycarbonate body was washed with laboratory detergent, rinsed with hot tap water followed by distilled water, and air dried at room temperature. Beetles were introduced singly into the Y-tube olfactometer at the entrance of the stem and allowed to move for 600 s. Thirty replications were carried out for each experiment. The insect behavior was measured in terms of residence time, i.e., the time spent by the beetles in each arm during the entire bioassay. For the experiments, each beetle was tested once and discarded if they did not make a choice (i.e., the beetles that did not enter in either of the two arms) over the observation period. After each bioassay, each individual was placed in a marked 2 mL vial, in order to be dissected for sex identification and female status (with eggs or without) under stereomicroscope (Zeiss Stereo Discovery V12). Overall, 270 *L. serricorne* adults were tested. Sex ratio was females 60.4% to males 39.6%. Twenty-six percent of the dissected females exhibited mature eggs.

The fruits used for the experiments were dried and reduced to a fine powder using an electric spice grinder. Powder (0.2 g) was used as a test stimulus in the glass jar while clean air was used as control.

### 2.3. Chemical Analysis of Capsicum Fruit Powder VOCs

To analyze the volatiles emitted from *C. annuum*, *C. frutescens* and *C. chinense*, two grams of powder of each species were separately placed in a 22 mL polytetrafluoroethylene silicon septum-lined cap (Supelco, Bellefonte, PA, USA) vial and subjected to head space solid phase micro-extraction (SPME) collection followed by GC-MS analysis. Three replications were carried out for each species. As internal standard for the analysis, 1 µL of a hexane solution containing 50 ng of nonadecane were placed into the vial. The SPME needle holder was then inserted through the septum and volatiles were absorbed on the exposed fiber for 60 min at 24 °C. The stationary phase used as coating fiber was polydimethylsiloxane (PDMS, 100 μm) (Supelco, Bellefonte, PA, USA). Fibers were conditioned in a gas chromatograph injector port as recommended by the manufacturer at 250 °C for 30 min. In order to perform the chemical analysis of the collected VOCs, the fibers were desorbed in the gas chromatograph inlet port for 2 min immediately after the end of the sampling time. Coupled gas chromatography-mass spectrometry (GC-MS) analyses of the headspace collections were performed on an Agilent 6890 GC system interfaced with an MS5973 quadruple mass spectrometer equipped with a DB5-MS column in spitless mode. Injector and detector temperatures were 260 °C and 280 °C respectively. Helium was used as the carrier gas. The GC oven temperature was set at 40 °C for 5 min, and then increased by 10 °C/min to 250 °C. Electron impact ionization spectra were obtained at 70 eV, recording mass spectra from 40 to 550 amu. The GC-MS peak area was calculated, and the relative quantitative composition estimated trough a comparison with the internal standard peak.

### 2.4. Bioassays with C. annuum Extracts and Fractions Thereof

In consideration of the results of the previous bioassays, in order to identify the active fraction of compound eliciting response to beetle adults, a second set of olfactometer bioassays was conducted with *C. annuum* extracts and the fraction thereof. *Capsicum annuum* extract was obtained by placing in a vial 2 g of fruit powder and 6 mL of hexane. After 10 min 3 mL of supernatant was collected and placed in a separated vial. Aliquots of 1 mL of extracts were fractionated by liquid chromatography into non-polar and polar fractions by using a solid phase extraction cartridge (SPE, silica column DSC-SI 52652-U, 1 mL tube, Supelco). The silica gel cartridge was conditioned by rinsing with 1 mL of hexane. The extract was loaded onto the cartridge as a hexane solution, followed by elution with 1 mL of hexane to obtain a non-polar hydrocarbons fraction. The cartridge then was eluted with 1 mL of dichloromethane to extract the polar compounds. The crude extracts and the fractions were kept at −20 °C until used for GC/MS analyses and behavioral bioassays. For the bioassays an aliquot of 600 µL of crude extract and/or extract fraction was placed in a filter paper cartridge 1 × 4 cm (Whatman, N°1). An equal amount of corresponding solvent was used as control. After 30 min for solvent evaporation, the test and control cartridges were placed in two separate 20 mL glass jars where the air was passing through.

### 2.5. Bioassays with α-Ionone and β-Ionone

In consideration of the results of the previous bioassays and of the chemical analysis of powder VOCs, a third set of olfactometer bioassays were conducted testing α-ionone and β-ionone as these molecules were the main compounds found in the attractant fraction (see result section). An aliquot of 10 µg of α-ionone or β-ionone dissolved in 200 µL, pipetted on a filter paper strip 1 × 4 cm (Whatman, N°1) was used as a test stimulus. These stimuli were tested versus 200 µL of hexane (negative control) or 600 µL of *C. annuum* crude extract (positive control). Cartridges were subject to solvent evaporation for 5 min before the start of an experiment.

### 2.6. Statistical Analysis

The residence time spent by the beetles in control and test arms was analyzed by *t*-test for dependent samples using Statistica 7.0 for Window (Statsoft 2001, Vigonza, PD, Italy).

## 3. Results

### 3.1. Bioassays with Capsicum Fruit Powders

Response of *L. serricorne* adults to volatiles from the three species of *Capsicum* powders are reported in Figure 1. Beetles were significantly attracted to volatiles of *C. annuum* fruit powder (t = 5.61; df = 29; *p* < 0.001; *N* = 30) and to *C. frutescens* (t = 2.52; df = 29; *p* = 0.018; *N* = 30). No significant attraction to *C. chinense* (t = 0.73; df = 29; *p* = 0.46; *N* = 30) was observed.

### 3.2. Chemical Analysis of Capsicum Fruit Powder VOCs

The main VOCs found in the SPME head space collections for the three species of *Capsicum* are reported in Table 1. Twenty-two different compounds were identified in the analysis. Eighteen compounds were identified from *C. frutescens* while 14 compounds were identified from *C. chinense* and *C. annum.* Qualitative and quantitative differences in the VOC profile of the three species were observed. In particular in *C. annuum* the five most abundant compounds were (mean percentage ± SE): dihydroactiniolide (19.52 ± 0.55), γ-himalachene (16.89 ± 1.17), β-ionone (16.46 ± 1.17), 2-methyl tetradecane (9.62 ± 0.75) and 2-methyl tetradecene (8.33 ± 0.98). In *C. frutescens* the five most abundant compounds were: dihydroactyniolide (12.65 ± 0.05), 2-methyl pentadecane (12.39 ± 0.07), 2-methyl tetradecane (11.58 ± 0.44), hexadecane (10.39 ± 37.14) and 2-methyl tetradecene (10.27 ± 0.33). In *C. chinense,* the five most abundant compounds were: dimethylcyclohexanol (47.41 ± 2.20), γ-himalachene (8.67 ± 0.18), 2-methyl tetradecane (6.87 ± 0.15), 2-methyl tetradecane (5.53 ± 1.10), and hexyl hexanoate (4.21 ± 1.70).

### 3.3. Bioassays with C. annuum Extracts and Fraction Thereof

*Capsicum annum* was selected to screen if the attractive compounds belonged to the polar or non-polar fraction. Responses of *L. serricorne* adults to volatiles from *C. annuum* extracts and fractions thereof are reported in Figure 2. Adults were attracted to *C. annuum* extract (t = 4.34; df = 29; *p* < 0.001; *N* = 30); and to the polar fraction (t = 8.69; df = 29; *p* < 0.001; *N* = 30) but were not attracted to the non-polar fraction (t = 1.53; df = 29; *p* = 0.13; *N* = 30).

### 3.4. Bioassays with α-ionone and β-ionone

The response of *L. serricorne* adults to α-ionone and β-ionone is reported in Figure 3. Beetles showed an attraction response toward both α-ionone (t = 4.02; df = 29; *p* < 0.01; *N* = 30) and β-ionone compared to hexane (t = 2.55; df = 29; *p* = 0.01; *N* = 30). Both compounds elicited a similar response in comparison with positive control (i.e., *C. annuum* crude extract), respectively: α-ionone (t = 0.11; df = 29; *p* = 0.91; *N* = 30) and β-ionone (t = 0.66; df = 29; *p* = 0.51; *N* = 30).

## 4. Discussion

The results demonstrate that *L. serricorne* is attracted by polar compounds emitted by *Capsicum* spp. powder; in particular, α-ionone and β-ionone seem to act as key mediators of this behavior. Insect preference towards *C. annuum* and *C. frutescens* rather than *C. chinense* was evident and qualitative and quantitative differences occurred in the VOCs profile of the three species. For example, α-ionone, β-ionone and dihydroactiniolide, main compounds of the polar fraction, were markedly more abundant in the VOCs of *C. annuum* and *C. frutescens* than in *C. chinense*.

The attraction of *L. serricorne* toward *C. annuum* fruit powder was already observed before [5] eliciting captures of males and females in pheromone trap experiments [9]. The bioassays conducted in this study with the extracts of *C. annuum* fruits established that only the polar fraction seems to mediate attraction behavior. Other studies have shown that other polar compounds may act as key odorants in attracting the cigarette beetle towards the substrate, e.g., catechol has been reported to attract *L. serricorne* adults toward coffee [35]. The *C. annuum* chemical analysis showed that among the polar compounds of fruit extract, the most abundant were α-ionone, β-ionone and dihydroactinidiolide. Further tests were conducted with the first two compounds due to their availability, cost and lesser toxicity. In fact, dihydroactinidiolide is reported to have a strong acetylcholinesterase inhibitory activity [36], is known to be a bioactive molecule in animals with cytotoxicity against human cells and harms soil and aquatic environments being toxic for several aquatic plants [36,37,38].

The bioassays using α-ionone and β-ionone individually showed that both attract adults, with a bioactivity similar to the *C. annuum* crude extract. α-Ionone and β-ionone are aroma compounds found in a variety of essential oils and present in the volatile blend of several plant species such as cotton and tomato [39,40]. Few studies report the behavioral function of α-ionone and β-ionone toward insects [41]. In particular, α-ionone has already been reported as an attractant for scarab pest *Macrodactylus subspinosus* F. (Coleoptera: Scarabaeidae) [42,43]. β-Ionone has been described as an attractant for the green leaf bug *Apolygus lucorum* (Meyer-Dür) (Heteroptera: Miridae) [44] and for males of *Euglossa mandibularis* (Hymenoptera: Apidae) [45]. Phoonan et al. [46] hypothesized that the attraction response of *L. serricorne* to dried leaves of blueberry was due by β-ionone. Observations of the current study support these findings suggesting that *L. serricorne* exploits such volatiles as key odorants to orient toward food and oviposition sources. The results previously showed by Guarino et al. [9] indicated that *C. annuum* fruit powder can increase the attraction response of *L. serricorne* adults to pheromone traps. Our results show that this response is mediated by α-ionone and β-ionone, molecules that could be used as attractants for future trap tests to implement pheromone captures or as attractants when used alone. Future efforts will focus on the evaluation of these molecules in specific trap tests conducted in food industries, storage facilities, museums, or herbaria infested by *L. serricorne* to evaluate their efficacy in increasing pheromone trap captures.

## 5. Conclusions

This study has identified α-ionone and β-ionone as key VOCs that attract *L. serricorne* adults to *C. annuum* dried fruits. These compounds can therefore be considered for *L. serricorne* management. Future tests will be conducted to determine the effect of these two molecules as attractants or pheromone-synergists in the field. The global economic importance of this pest justifies the continuation of research supporting the identification of new molecules and technologies for creating new management methods.

## Figures and Tables

**Figure 1 insects-12-00061-f001:**
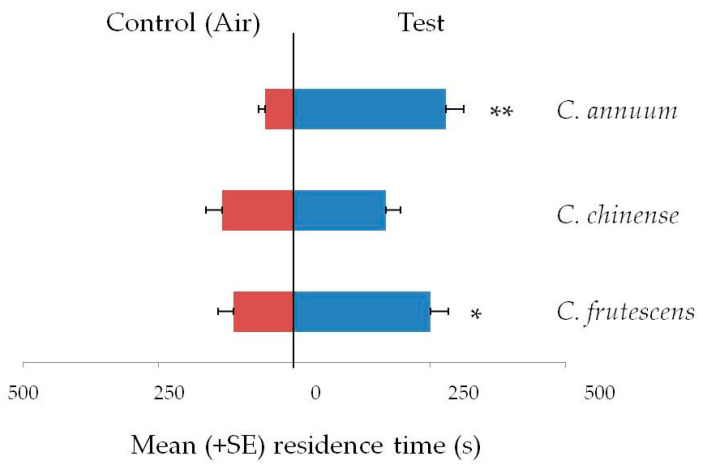
Olfactory response of *Lasioderma serricorne* to volatiles of dried fruit powder of three *Capsicum species*, *C. annuum*, *C. chinense* and *C. frutescens*, in Y-tube olfactometer. Bars indicate the mean residence time (+SE) adult beetles spent in the arm with the odor source. Thirty insects tested per treatment. *T*-test for dependent samples, * *p* < 0.05 ** *p* < 0.01.

**Figure 2 insects-12-00061-f002:**
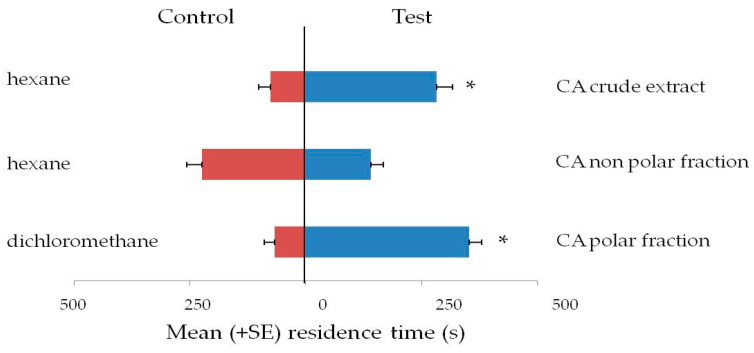
Olfactory response of *Lasioderma serricorne* adults to *C. annuum* (CA) extracts and fractions thereof in a Y-tube olfactometer. Bars indicate the mean residence time (+ SE) adult beetles spent in the arm with the odor source. Thirty insects tested per treatment, *T*-test for dependent samples, * *p* < 0.01.

**Figure 3 insects-12-00061-f003:**
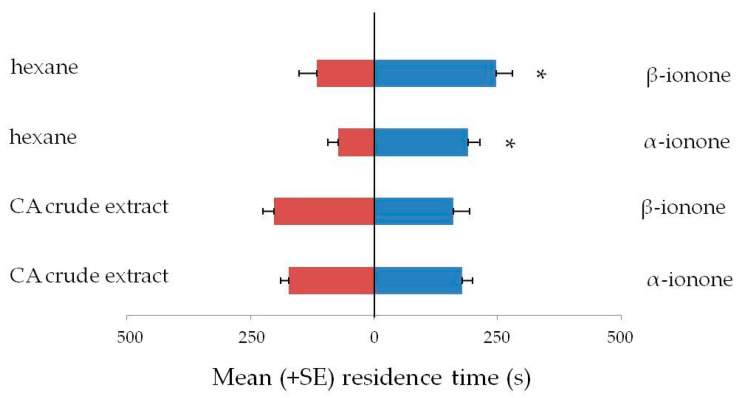
Olfactory response of *Lasioderma serricorne* adults to α-ionone and β-ionone versus solvent or *C. annuum* extract in Y-tube olfactometer. Bars indicate the mean residence time (+SE) adult beetles spent in the arm with the odor source. Thirty insects tested per treatment, *t*-test for dependent samples, * *p* ≤ 0.01.

**Table 1 insects-12-00061-t001:** Chemical composition of the VOC (ng) profile obtained from the HS-SPME air collection from the fruit powders of *Capsicum annuum*, *C. chinense* and *C. frutescens* Mean (±SE) emission rates are reported in ng, though comparison with 50 ng of nonadecane used as internal standard.

Chemical	Polarity	RT (min)	RI (DB5MS)	*C. anuum*	*C. chinense*	*C. frutescens*
2-methyl tridecane	NP	16.804	1364	0.00	0.00	76.52 ± 23.59
hexyl hexanoate	P	17.147	1389	0.00	40.90 ± 26.27	0.00
dimethylcyclohexanol	P	17.246	1396	0.00	374.51 ± 81.39	0.00
tetradecane ^a^	NP	17.294	1400	0.00	21.59 ± 12.72	0.00
α-ionone ^a^	P	17.655	1428	62.85 ± 37.18	8.27 ± 6.41	19.08 ± 1.82
caryophyllene ^a^	NP	17.714	1435	137.65 ± 51.27	0.00	0.00
2-methyl tetradecene	NP	17.883	1446	161.28 ± 72.69	48.97 ± 22.66	127.27 ± 32.89
2-methyl tetradecane	NP	18.123	1464	164.44 ± 39.90	55.28 ± 13.78	147.86 ± 45.33
β-ionone ^a^	P	18.388	1484	289.81 ± 84.55	42.47 ± 8.66	87.70 ± 26.33
germacrene d	NP	18.481	1492	80.38 ± 24.94	32.72 ± 8.35	13.33 ± 1.59
γ-himalachene	NP	18.525	1495	290.02 ± 73.05	68.27 ± 15.07	6.82 ± 1.48
pentadecane ^a^	NP	18.584	1500	71.40 ± 26.03	36.44 ± 5.54	57.29 ± 15.53
muurolene	NP	18.616	1502	30.05 ± 5.66	27.91 ± 5.95	0.00
cadinene	NP	18.691	1509	27.01 ± 9.07	19.22 ± 4.11	0.00
dihydroactyniolide	P	19.105	1543	354.13 ± 123.42	32.33 ± 8.29	158.17 ± 42.63
2-methyl pentadecane	NP	19.364	1564	80.24 ± 27.73	0.00	154.98 ± 42.18
3-methyl pentadecane	NP	19.451	1572	0.00	0.00	18.85 ± 5.27
z 3-hexadecene	NP	19.603	1584	0.00	0.00	41.26 ± 11.19
hexadecane ^a^	NP	19.801	1600	0.00	0.00	133.39 ± 37.14
8-heptadecene	NP	20.316	1645	0.00	0.00	36.54 ± 10.27
2-methyl hexadecane	NP	20.537	1664	0.00	0.00	96.98 ± 25.62
heptadecane ^a^	NP	20.950	1700	49.45 ± 20.97	0.00	84.35 ± 18.76
octadecane ^a^	NP	22.046	1800	0.00	0.00	11.43 ± 2.93

RT = retention time, RI = experimentally determined retention index (DB5MS column), ^a^ = compound identified by authentic standard, P = compound belonging to the polar fraction, NP = compound belonging to the non-polar fraction.

## Data Availability

All data generated or analyzed during this study are included in this published article.

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
