# Peer review of "Odorants of *Capsicum* spp. Dried Fruits as Candidate Attractants for *Lasioderma serricorne* F. (Coleoptera: Anobiidae)"

_insects, 2021, doi:10.3390/insects12010061_

Round 1

Reviewer 1 Report

This study identified the volatile attractants of chili fruits for the stored product pest Lasioderma serricorne for a potential application in bait traps. They found differential attractiveness of three different chili species that correlated with abundance of multiple compounds in the volatile bouquet of these chili species, especially three compounds of the polar phase from a hexane extract, alpha and beta ionone as well as dihydroactinidiolide. A further test of alpha and beta ionone proved that both are attractive for L. serricorne in comparison to a solvent control. Dihydroactinidiolide was not considered due to its reported toxicity and thus complicated inclusion in traps.

In general I find the study well conducted and written, , but have a few minor questions that should be addressed before acceptance.

Could you please specify the source of the Lasioderma serricorne culture and chili fruits in the method section, not only the acknowledgements.

For the bioassays with extracts and pure compounds: Could you know the relation of the single alpha and beta ionone amounts to the (estimated) amount in the fruit extracts? Where similar or much higher concentrations used?

Author Response

Dear reviewer,

thank you for your useful suggestions and corrections. We accept all your comments, as we include the source of the Lasioderma serricorne culture and chili fruits in the method section. Regarding alpha and beta ionone amounts in the fruit extracts, we could not quantify directly but we estimated the amount to use based for the further bioassay on our experiences on semochemicals. Our choice of 10 micrograms of tested compounds was arbitrary, in order to evaluate the attraction response. The conditions of headspace SPME collection (static) and olfactometer bioassay (dynamic) were different, so it is hardly to compare the two amounts. However our choice was based also on practical consideration for further applications in the traps.

Thank you again

Reviewer 2 Report

Guarino et al.

This is an interesting contribution on the use of chili as additional attractants for Lasioderma serricorne. In particular, the authors test chili powders from Capsicum annuum, C. frutescens, and C. chinense. The authors tested 30 CB adults, and found significant attraction in a Y-tube assay to two of these species. The authors then characterized the volatile composition from each using SPME coupled with GC-MS. Finally, the behavioral response of CB was determined to crude extracts of polar and nonpolar fractions of the chili powders, as well as to the most common volatiles, alpha-ionone and beta-ionone. The authors found significant attraction, evidenced by increased residence time. Before recommending acceptance, I have the clarifying questions and comments below. I believe the contribution needs moderate language revision—I provide some minor changes below for the abstract and first paragraph of the introduction.

Points of clarification

  • Lines 29-31, the authors do not test chili powders in combination with existing pheromone technology, so there is nothing being synergized in this paper. Rephrase or delete this sentence.
  • Why don’t the authors show the proportion of individuals choosing each stimulus? This is generally and more commonly provided along with a measure of retention.
  • Line 58-60 not sure what the authors mean here, please rephrase for clarity
  • Why didn’t the authors try to test a mixture of the most common compounds? Prior research has shown that mixtures do better for attraction: https://onlinelibrary.wiley.com/doi/full/10.1111/j.1570-7458.2009.00954.x
  • Section 2.3, the authors don’t specify how many volatile replicate samples they ran using SPME/GC-MS. This needs to be included.
  • On Table 1: it would be helpful if the authors include an additional row for each species and indicate the % contribution to the total headspace emission for each compound.
  • 254-255 “Blueberry leaves tea” – what is this supposed to be? Tea plant is a separate species from blueberry plants.
  • 268-270 The authors need to delete this sentence from the conclusion—they did not test anything having to do with trapping or mass trapping, so it’s odd that it makes an appearance here when they have no data to support this.
  • Throughout the MS, the authors refer to chili powder as “chili dried fruits powder”—but it would suffice to refer to it simply as “chili powder”. This is the standard expression for this. It is otherwise awkward, unwieldly, and not idiomatically correct.

Minor comments:

Line 29 damage is always singular in English

Line 29 replace “as” with “at”

Line 30 rephrase as “can be positively”

Line 32 see comment above, delete “fruits”

Line 47 change pest to pests and rephrase to “in the food storage”

Line 48 change “damages” to “damage” and “in” to “at”, insert “such” before “as museums”

Line 51 delete “the” change “determine” to “inflicts”

Line 52 change “excrements” to “frass” and “gnawed particles” to “damage”

Line 57 rephrase to “lead to a holistic integrated pest management (IPM) program, which is strongly…”

Line 63 change “that” to “which” and delete “the” before “male” and insert s on the end of “beetle”.

Line 72 delete “dried fruits”

Line 73 rephrase as “resulted in attraction under laboratory conditions”

Line 97 delete s from “beetles” and delete “dried fruits of”

Line 98 insert a colon prior to the species names.

Line 114 change “response” to “choice”

Line 115 replace “neither” with “either”

Line 116 insert space between 2 and ml

Line 119 insert “were” after “males” and delete “the” before percentage, change “presented” to “exhibited”

Line 225 delete “fruit”

Line 227 delete “some species of Capsicum as”

Line 228 change “evidenced” to “supported”

Line 243 change “pond weeds” (this is a pejorative and subjective phrase) to “aquatic plants”

Line 246 change “evidenced” to “showed”

Author Response

Dear Reviewer 2,

Thank you for your letter and the opportunity to improve our manuscript with your valuable comments and corrections. We have included all yours comments and responded to them individually, indicating exactly how we addressed each concern or problem and describing the changes we have made (in red colour in the main text).

Please see below the points of clarification and our answer.

  • Lines 29-31, the authors do not test chili powders in combination with existing pheromone technology, so there is nothing being synergized in this paper. Rephrase or delete this sentence.

RESPONSE: According with the Reviewer suggestion, we rephrased the sentence.

  • Why don’t the authors show the proportion of individuals choosing each stimulus? This is generally and more commonly provided along with a measure of retention.

RESPONSE: in consideration of the instrument used and of the high walking activity of L. serricorne we preferred to use the parameter “residence time” in test and control arms that evidence the dynamic response of the beetle.

  • Line 58-60 not sure what the authors mean here, please rephrase for clarity

RESPONSE: According with the Reviewer suggestion, we rephrased the sentence.

  • Why didn’t the authors try to test a mixture of the most common compounds? Prior research has shown that mixtures do better for attraction: https://onlinelibrary.wiley.com/doi/full/10.1111/j.1570-7458.2009.00954.x

RESPONSE: Indeed the Reviewer comment is correct. However, we decided to test the molecules singly in the perspective to identify the key odorants eliciting attraction response, acting similarly to the crude extract as shown in the results. In addition we consider the compound that could be easily used for further applications in traps being no toxic and more available in the market and easily to extract.

  • Section 2.3, the authors don’t specify how many volatile replicate samples they ran using SPME/GC-MS. This needs to be included.

RESPONSE: This was our oversight. According with the Reviewer suggestion, we added this information in section 2.3.

  • On Table 1: it would be helpful if the authors include an additional row for each species and indicate the % contribution to the total headspace emission for each compound.

RESPONE: We appreciate the comment from the reviewer. However, in consideration that including other three columns in the table, that would make less readably, we decided to report the percentages of the five most abundant compound of each species directly in the text.

  • 254-255 “Blueberry leaves tea” – what is this supposed to be? Tea plant is a separate species from blueberry plants.

RESPONSE: We corrected this oversight.

  • 268-270 The authors need to delete this sentence from the conclusion—they did not test anything having to do with trapping or mass trapping, so it’s odd that it makes an appearance here when they have no data to support this.

RESPONSE: As suggested from the reviewer we deleted this sentence from the conclusion.

  • Throughout the MS, the authors refer to chili powder as “chili dried fruits powder”—but it would suffice to refer to it simply as “chili powder”. This is the standard expression for this. It is otherwise awkward, unwieldly, and not idiomatically correct.

RESPONSE: thank you for that comment we correct it throughout the manuscript.

Answee to minor comments:

Line 29 damage is always singular in English - done

Line 29 replace “as” with “at” - done

Line 30 rephrase as “can be positively” - done

Line 32 see comment above, delete “fruits” - done

Line 47 change pest to pests and rephrase to “in the food storage” - done

Line 48 change “damages” to “damage” and “in” to “at”, insert “such” before “as museums” - done

Line 51 delete “the” change “determine” to “inflicts” - done

Line 52 change “excrements” to “frass” and “gnawed particles” to “damage” - done

Line 57 rephrase to “lead to a holistic integrated pest management (IPM) program, which is strongly…” - done

Line 63 change “that” to “which” and delete “the” before “male” and insert s on the end of “beetle”. - done

Line 72 delete “dried fruits”- done

Line 73 rephrase as “resulted in attraction under laboratory conditions” - done

Line 97 delete s from “beetles” and delete “dried fruits of”- done

Line 98 insert a colon prior to the species names.

Line 114 change “response” to “choice”- done

Line 115 replace “neither” with “either”- done

Line 116 insert space between 2 and ml

Line 119 insert “were” after “males” and delete “the” before percentage, change “presented” to “exhibited” - done

Line 225 delete “fruit” - done

Line 227 delete “some species of Capsicum as”- done

Line 228 change “evidenced” to “supported” - done

Line 243 change “pond weeds” (this is a pejorative and subjective phrase) to “aquatic plants” - done

Line 246 change “evidenced” to “showed” – done

Reviewer 3 Report

Dear Authors:

The subject of the manuscript is interesting, but the English text in all sections must be improved. Please collaborate with a colleague with English as his/her mother tongue to revise this manuscript. Be careful to use past tense in a sentence, not past and present tenses. Use chili fruit powders, rather than chili fruits powders. The other examples are too numerous to list. 

Author Response

Dear Reviewer 3,

Thank you for your comments, as you suggested the manuscript was revised by a mother tongue English colleague as indicated in the acknowledgements.

Round 2

Reviewer 3 Report

Dear Authors:

Please conduct a minor revision of your interesting manuscript following the suggestions below.

L19: to the volatile organic compounds (VOCs)

L28: and is increasingly impacting museums...

L30: to the volatile organic compounds (VOCs)

L62-66: missing references 28, 29, 30.

L71: L. serricorne in laboratory studies

L89: Authors, please define 00 flour and 0 flour.

L95: Please provide authority names for species mentioned for the first time in the text.

L116: Twenty-six percent of the...

L121: two grams of powder

L143: Capsicum anuum extract was obtained...

L170: Figure 1

L171-172: Include df (degrees of freedom) values

L180: Table 1

L201-203: Include df values

L210-213: Include df values

L220-222: The results demonstrate.... Please avoid single sentence paragraphs.

L234: Further tests were conducted with the first two compounds...

L245: the green leaf bug...

L247: leaves of blueberry was due to B-ionone.

L250: fruit powder can increase the...

L252: could be used as attractants...

---

Author Response

Dear Reviewer 3,

thank you for your comments that improve our manuscript and that we fully accept. We have indicated in blue that changes made according to your suggestions.

Please accept our best wishes for the New year.

Sincerely

Barbara